# COVID-19 and Lung Cancer: A Comprehensive Overview from Outbreak to Recovery

**DOI:** 10.3390/biomedicines10040776

**Published:** 2022-03-26

**Authors:** Maristella Bungaro, Francesco Passiglia, Giorgio V. Scagliotti

**Affiliations:** Department of Oncology, University of Turin, San Luigi Hospital, 10043 Orbassano, Italy; maristellabng@gmail.com (M.B.); giorgio.scagliotti@unito.it (G.V.S.)

**Keywords:** SARS-CoV-2, COVID-19, lung cancer

## Abstract

Lung cancer patients have been associated with an increased risk of COVID-19 infection, pulmonary complications, and worse survival outcomes compared to the general population. The world’s leading professional organizations provided new recommendations for the diagnosis, treatment, and follow-up of lung cancer patients during the pandemic as a guide for prioritizing cancer care issues. Telemedicine was preferred for non-urgent consultations, and screening programs were temporarily suspended, leading to possible diagnostic delays along with an estimated increase in cause-specific mortality. A vaccine campaign has recently emerged as the main weapon to fight the COVID-19 pandemic, inverting this negative trend. This work aims to provide a comprehensive overview of the epidemiology and immune-pathophysiology of SARS-CoV-2 infection in cancer patients, highlighting the most relevant changes in the clinical management of lung cancer patients during the pandemic.

## 1. Epidemiology, Transmission and Clinical Features of SARS-CoV-2 Infection

In December 2019, a pneumonia cluster of unknown etiology appeared in the city of Wuhan (China’s Hubei). Virus isolation and molecular analysis allowed a novel coronavirus (CoV) to be identified, named SARS-CoV-2, by the International Committee on Virus Taxonomy, representing the seventh member of the Coronaviridae family to infect humans. Initial investigations suggested that it could be originally transmitted by bats, as SARS CoV-2 had a nucleotide identity that was a 96% match with bat coronavirus, although the real origin of the infection is still debated [1].

The main clinical features of SARS CoV-2 disease (COVID-19) are fever, cough, muscle pain, and dyspnea, while atypical symptoms include diarrhea and vomiting. The clinical severity of COVID-19 was graded as follows: asymptomatic disease (positive SARS CoV-2 PCR test without clinical signs of infection), mild disease (symptoms of acute upper respiratory tract infection without pneumonia), moderate disease (radiological evidence of pneumonia), severe disease (pneumonia with dyspnea and hypoxemia), and critical disease (pneumonia with acute respiratory distress syndrome, respiratory failure, shock, and multiple organ dysfunction) [2]. More than 80% of COVID-19 cases in adults were classified as mild/moderate, while the majority of infections in children are asymptomatic [2]. Conversely, patients with comorbidities (age ≥ 65 years, smoking habits, chronic cardiovascular and pulmonary diseases, renal insufficiency, sickle cell disease, diabetes, obesity, pregnancy, and cancer) have been associated with a higher risk of progressing to the severe/critical disease stages [3].

The known routes of COVID-19 transmission were droplets as well as direct physical contact, while the possibility of fecal transmission has not been confirmed yet [4]. Vertical transmission has been described only in a small subgroup of cases and has been mainly related to the third trimester of pregnancy [5]. Infected individuals may be contagious before the onset of symptoms [6]. The estimated incubation period is up to 14 days from the time of exposure, with a median incubation ranging from four to five days [6]. Asymptomatic infection, particularly in children, is an important source of disease for the community, as these asymptomatic patients can easily cause familial clusters.

Considering the dramatic increase of the number of COVID-19 confirmed cases, the World Health Organization initially declared this epidemic a public health emergency of international concern on the 30 January 2020, while the COVID-19 disease was definitively declared a pandemic on the 11 March 2020. To date, more than 240 million cases of COVID-19 and about 4.9 million deaths have been reported worldwide [7].

## 2. Immune-Pathophysiology of SARS-CoV-2 Lung Injury and Risk of Infection in Lung Cancer Patients

Cancer patients have been associated with an increased risk of COVID-19 infection compared to the general population, due to the systemic immunosuppression caused by both the tumor itself and the anti-cancer treatments [8]. Specifically, lung cancer patients may have an increased risk of pulmonary complications from COVID-19 (such as admission to the intensive care unit for invasive ventilation) with worse survival outcomes [9,10]. A descriptive analysis of patients entering the hospital emergency department for symptoms related to SARS-CoV-2 infection showed that COVID-positive cancer patients were older, harboring ≥ 2 comorbidities, more frequently developing respiratory failure, at higher rates of hospitalization, compared to the general population [11]. Liang et al., in collaboration with the National Health Commission of the People’s Republic of China, identified a prospective cohort of patients hospitalized following the diagnosis of COVID-19 disease. Just 1% of these patients had a history of cancer, with lung cancer accounting for 5 out of 18 cancer patients [12]. A retrospective analysis of 1524 oncological patients by Yu et al. showed an increased risk of SARS-CoV-2 infection (odds ratio (OR), 2.31; 95% confidence intervals (CI), 1.89 to 3.02) compared with the general population [13]. The risk appeared to be increased both in patients with and without active anticancer treatment, with non-small cell lung cancer (NSCLC) patients aging ≥ 60 years old being the most likely to develop COVID-19 [13].

Severe COVID-19 can be considered a hyperinflammatory disorder characterized by a massive activation of the immune system, thus explaining the worse survival outcomes observed in both elderly people and cancer patients. The pathophysiology of COVID-19 has not been entirely clarified yet. In some cases, SARS-CoV-2 induces an excessive and aberrant ineffective host immune response resulting in a severe and potentially fatal lung injury [14]. In severe cases, infection may be associated with the hyperactivation of tissue macrophages that release a storm of cytokines leading to rapidly progressive organ dysfunction. Macrophage activation syndrome (MAS) can be fatal due to pancytopenia, tissue hemophagocytosis, disseminated intravascular coagulation, and hepatobiliary and central nervous system dysfunction [15].

Qin and colleagues investigated the immune response of 452 patients with COVID-19 and reported an increased neutrophil/lymphocyte ratio (NLR) and T lymphopenia, which were more pronounced in severe disease than in mild disease [14]. Patients with severe COVID-19 also reported higher serum levels of pro-inflammatory cytokines (TNF-α, IL-1 and IL-6) and chemokines (IL-8), suggesting a possible role for hyper-inflammatory responses in the pathogenesis of COVID-19 disease [14]. The differentiation of naïve CD4+ T cells into effector and memory cells as well as the balance between these elements is crucial for maintaining an efficient cell-mediated immune response [16]. Patients with severe disease seem to have a dysregulated immune system, with a higher ratio of naive cells to memory cells and a decreased number of regulatory T cells [14]. Regulatory T cells play a crucial role in regulating the activity of a wide range of immune cells to maintain self-tolerance and immune homeostasis [17]. Both helper T and suppressor T cells in patients with COVID-19 were decreased, but a lower level of helper T cells was found in the severe group [14]. An increased expression of pro-inflammatory cytokines and chemokines along with a consumption of CD4+ and CD8+ T cells, could result in severe inflammatory responses in COVID-19 patients. A detailed characterization of natural killer (NK) cells in COVID-19 patients reported elevated plasma levels of interferon (IFN)-α in severe disease resulting in an intense and prolonged IFN-α-induced NK cell response [18]. NK cells appeared to also play a specific role in the development of lung fibrosis for those patients harboring severe COVID-19 disease, as they had impaired antifibrotic activity [18].

Following the characterization of the immune response to SARS-CoV-2 infection, a series of immune-biological biomarkers have been investigated for predicting the outcomes of patients who develop COVID-19 disease. A recent study by Del Valle and colleagues showed that elevated levels of immune-biomarkers involved in the cytokine storm induced by COVID-19 at the time of hospitalization, like both IL-6 and TNF-α, were significantly associated with patients’ survival [19].

Biomarker studies coming from Gustave Roussy Cancer Centre cohort of cancer patients managed for COVID-19 found that high levels of C-reactive protein (PCR) and lactate dehydrogenase (LDH) were associated with the disease severity, while levels of PCR and D-dimer predicted an increased risk of death from COVID-19 disease [20].

Tian et al. demonstrated that pro-inflammatory biomarkers, including TNF-α, IL-6, procalcitonin, and PCR, as well as levels of leukocytes, neutrophils, LDH, coagulation factors and NT-proBNP were significantly associated with an increased severity of COVID-19 disease in cancer patients admitted to a hospital department [21]. In contrast, both albumin levels and albumin-to-globulin ratio were associated with lower disease severity. The TNF-α, NT-proBNP, albumin-to-globulin ratio, and CD4+ T-cells were also associated with an increased risk of death from COVID-19 [21]. In patients with non-severe disease, CD4+ T cells decreased during the first 3 weeks of hospitalization and then progressively increased. The decrease in CD4+ T cells among patients with severe COVID-19 has shown to be more pronounced and prolonged over time [21].

The inflammatory response, along with endothelial dysfunction and microvascular damage, may underly the effects of COVID-19 infection, including pulmonary fibrosis and subsequent reduced respiratory function [22]. It was observed that 3 months later an acute infection, some patients had CT abnormalities including ground glass opacities (GGO) and subpleural bands [23]. Conversely some patients experienced GGOs resolution at six months while developing fibrosis with or without parenchymal distortion [23]. Pulmonary function tests showed a decreased forced vital capacity (FVC), total lung capacity (TLC), and carbon monoxide diffusing capacity (DLCO) < 80% [22]. Coronavirus infection could directly promote pulmonary fibrosis through two different mechanisms. The nucleocapsid protein of SARS-CoV-1, more than 90% similar to that of SARS-CoV-2, increases levels of transforming growth factor-beta (TGF-β), which acts as a potent promoter of pulmonary fibrosis. Coronaviruses also appear to induce downregulation of the angiotensin-converting-enzyme-2, reducing angiotensin II clearance in the lungs. Angiotensin II may in turn upregulate TGF-β and connective tissue growth factor [22]. Post infection pulmonary fibrosis can be estimated at 2–6% after moderate disease [24]. One year later, the incidence rate of impaired DLCO and persistent radiological lung damage still exceeds 30% [24]. Despite the efforts of the worldwide medical community, there are no treatment options for COVID19-induced pulmonary fibrosis.

Recent studies investigated the critical role of reactive oxygen species (ROS)-associated inflammation pathways (in particular the redox-sensitive transcription factor NRF2) in both COVID-19 and lung cancer, showing some similarities but relevant differences [25]. Indeed, NRF2 is usually activated in lung cancer, facilitating the immune escape of tumor cells, while it is downregulated in COVID-19-positive patients, causing immunosuppressive effects that may worsen COVID-19 symptoms in lung cancer patients [25].

Westblade and colleagues conducted a multicenter study including more than 3000 hospitalized COVID-19-positive patients with the aim of demonstrating the correlation between SARS-CoV-2 viral load and COVID-19 mortality [26]. Results showed a mortality rate of 38.8%, 24.1%, and 15.3% among patients with a high, medium, and low viral load, respectively [26]. Similar results have also been observed in cancer patients, suggesting that the SARS-CoV-2 viral load on admission to hospital could be highly predictive of cause-specific mortality in both patients with and without cancer [26].

A recent retrospective analysis by Luo et al. reported a longer and more severe course of COVID-19 disease in lung cancer patients than in the general United States (US) population, which is consistent with most recent literature data [27]. Both smoking habits and chronic obstructive pulmonary disease were found to be associated with infection severity, whereas tumor characteristics and anticancer treatments did not influence patients’ symptoms. Despite the burden of COVID-19 disease in lung cancer patients, more than half of them achieved recovery, including those who initially required invasive ventilation. Interestingly, only a small fraction (11%) of the overall deaths occurring in lung cancer patients during the pandemic could be ascribed to COVID-19 [27]. The multicenter observational TERAVOLT study showed a COVID-19 mortality rate of 33% along with a 76% hospitalization rate in patients with pleuropulmonary neoplasms [28]. Among the hospitalized patients, 88% met the criteria for intensive care unit admission, but only 10% had access to the intensive care unit. In line with previous data, cigarette smoking has been confirmed to be significantly associated with patients’ mortality in multivariate analysis, whereas the type of systemic anticancer therapy did not influence the survival outcomes of COVID-19-positive patients [28].

Specific anti-viral drugs were administered to hospitalized patients affected by severe forms of infection, including also cancer patients undergoing active antineoplastic treatments. A recent systematic review of the literature did not find a high rate of interactions [29], reporting cardiological side effects with chloroquine/hydroxychloroquine and chemotherapy or trastuzumab [29]. Tocilizumab seems to interfere with the activity of several checkpoint inhibitors, decreasing the concentration of some tyrosine kinase inhibitors (TKIs) (ceritinib, crizotinib, brigatinib, and gefitinib) and docetaxel [29]. Finally, literature currently provides an inconclusive picture of potential interactions between anti-viral drugs and antitumor therapies in cancer patients.

## 3. Management of Lung Cancer Patients’ Care during the COVID-19 Pandemic

Since the beginning of the pandemic, the oncology community has been pushed to find a balance between protecting cancer patients from the risk of infection and ensuring adequate anti-cancer treatment. In order to comply with social distancing and general public health measures to mitigate the spread of SARS-CoV-2, outpatient oncology services have been thoroughly reorganized. Triage areas have been set up at the entrance of the hospitals in order to administer COVID-19-related symptom questionnaires and provide body temperature checks [30]. Non-essential outpatient visits were postponed or performed via telemedicine (e.g., long-term follow-up visits in surgically resected patients with low risk of relapse). For some patients on active treatment, whenever possible, blood tests were performed at home.

Several leading global professional organizations, including the European Society of Medical Oncology (ESMO), provided recommendations for the diagnosis, treatment, and follow-up of lung cancer patients during the COVID-19 pandemic, as a guide for prioritizing cancer care issues and mitigating potential harm related to the state of health emergency [31]. The proposed recommendations considered three levels of priority (high, medium, and low) for therapeutic interventions, according to the Cancer Care Ontario criteria, Huntsman Cancer Institute, and Magnitude of Clinical Benefit Scale, taking into account patients’ clinical stability, the magnitude of benefit in terms of survival, quality of life, or both, and the negative impact that delayed treatment might have on the overall outcomes [32,33].

Clinical decision making within a multidisciplinary setting has been strongly recommended for a multifactorial risk/benefit assessment, including the extent of the outbreak in the country, the resources of the local health facility, and the risk of individual infection.

Patients with a high suspicion of newly diagnosed lung cancer should be managed within standard diagnostic pathways, without delaying radiological imaging as well as the rest of the diagnostic work-up.

It has been necessary to prioritize early stage disease requiring surgery among the different surgical procedures, while maintaining the highest possible standards, even if both surgical activities and ICU access have been dramatically limited. Considering the risk of SARS-CoV-2 infection in surgically resected patients and the potential immunosuppressive status induced by peri-operative chemotherapy, it has been recommended that the role of adjuvant treatment be reconsidered following thorough discussion with individual patients. The indication should be always denied in frail, elderly patients, who are affected by significant comorbidities.

The clinical management of stage III NSCLC has been particularly challenging during the COVID-19 pandemic, considering the need to optimize timing and sequencing of chemoradiation without increasing the risk of exposure to SARS-CoV-2. Given its curative potential, the treatment of stage III NSCLC patients maintained the highest priority, considering a delayed 4 week interval for durvalumab consolidation infusions, where permitted by national regulatory agencies.

As for newly diagnosed metastatic disease, it has been suggested that potential alternative administration schedules for the immune checkpoint inhibitors be evaluated, using either nivolumab every 4 weeks or pembrolizumab every 6 weeks, where permitted, in order to reduce patients’ access to the oncological day hospitals [34,35]. A home delivery service was also recommended for patients receiving either tyrosine kinase inhibitors (TKIs) or oral chemotherapy, while venous or subcutaneous antiresorptive bone-protective therapy should be discontinued. Second-line therapies should be carefully evaluated at single patient level.

Treatment of small cell lung cancer (SCLC) remained a priority while prophylactic cranial irradiation (PCI) should be potentially postponed in patients with limited stage and replaced by MRI surveillance in patients with extensive SCLC. The main ESMO recommendations for the clinical management of patients with lung cancer in the COVID-19 era are summarized in Table 1.

A survey of thoracic oncologists during the first wave of the coronavirus disease outbreak in Italy reported that although most clinicians followed national and international treatment guidelines in the adjuvant and first-line setting, 77% of them reported significant changes in the management of outpatients, especially in the selection of second or further lines of treatment [36].

An American Society for Radiation Oncology (ASTRO)-European Society for Radiation Oncology ESTRO consensus conference of 32 experts in lung cancer radiotherapy produced some practical recommendations on radiotherapy treatment delivery, considering two different pandemic scenarios: mitigation of infectious risk versus reduced radiotherapy resources [37]. The first expert recommendation aimed to not compromise patients’ prognosis by deviating from recommended practice guidelines. Secondly, radiotherapy postponement or discontinuation for COVID-19-positive patients should always be considered to avoid the exposure of negative patients and health care personnel to a risk of infection. In a severe pandemic scenario with limited resources, it has been recommended that patients’ triage considering their cure potential, treatment benefit, life expectancy, and performance status [37].

## 4. Effect of COVID-19 on Lung Cancer Diagnosis and Treatment Delays

Quantifying the impact of COVID-19 on cancer diagnosis and treatment delays remains an appealing and challenging topic. Elliss-Brookes et al. showed that cancer diagnostic delays of 2 weeks are associated with a later stage of disease at presentation [38]. Furthermore, diagnosis after an initial admission to an emergency department appears to be associated with worse survival outcomes than all other routes [38]. A recent national population-based modeling study estimated the number of cancer deaths attributable to delays in cancer diagnosis as a result of COVID-19 blockade in the United Kingdom (UK) [39]. The estimated increase in cancer deaths up to 5 years after diagnosis, ranges from 4–8% for lung cancer to 16% for colorectal cancer [39]. A retrospective single-center study conducted at the National Hospital Organization Kyoto Medical Center reported a lung cancer treatment delay rate of 9.1% during the COVID-19 pandemic [40]. Most patients were treated with immunotherapy and delayed administration at their own request. No TKIs-treated patients experienced treatment delays [40]. A recent Italian study presented at the World Conference on Lung Cancer (WCLC) 2021, reported a 4.4% delay in diagnostic, therapeutic, and palliative oncological procedures [41]. A total of 135 procedures were performed in 125 lung cancer patients, most including lung biopsies with diagnostic intent. A nasopharyngeal swab for SARS-CoV-2 screening was performed before each procedure to avoid the spread of the virus within the hospital environments. Six procedures were postponed due to COVID-19 positivity and were performed at recovery with a median delay of 36 days (range 14–55) [41]. The COVID-19 pandemic also required an unprecedented disruption to cancer screening programs. An expert panel of 24 members, including pulmonologists, thoracic radiologists, and thoracic surgeons was formed to provide consensus statements on lung cancer screening and lung nodule evaluation during the pandemic (CHEST Expert Panel Report) [42]. The consensus on delaying the evaluation of lung nodules with indolent features incidentally detected by screening was unanimous. The consensus was less uniform as regards the management of nodules with a mean diameter > 8 mm: if the probability of malignancy is greater than 85%, such nodules should not undergo further diagnostic tests and the treatment approach should be discussed directly within a multidisciplinary setting [42]. The University of Cincinnati oncology group reported the results of their low-dose computed tomography (CT) screening program, comparing a pre-COVID19 baseline period with the pandemic period [43]. Screening was suspended between March and June 2020. There was a total decrease in terms of monthly consultations as well as in the number of new patients screened during the COVID-19 period. The proportion of patients with suspected malignant pulmonary nodules (Lung-RADS 4) was significantly increased after resumption of screening (29% vs. 8%), and a decrease in visit compliance was also observed after reopening the program [43].

## 5. Effect of Anticancer Treatment on COVID-19 Cancer Patients

Some studies suggested that COVID-19-positive patients receiving systemic anticancer therapy were at higher risk of reporting severe clinical outcomes than patients not receiving any anticancer treatment [44], while other studies did not support this hypothesis [45,46]. Two recent meta-analyses were conducted to clarify whether and which anticancer treatments may produce a detrimental effect on clinical outcomes in COVID-19-positive cancer patients.

Liu et al. analyzed data from 29 studies involving more than 5000 patients undergoing different anticancer treatments [47]. The most common type of treatment among cancer patients with COVID-19 was chemotherapy (30% pooled rate), followed by targeted therapy, radiotherapy, endocrine therapy, surgery, and immunotherapy. There were no significant differences in terms of mortality between patients who received or did not receive anticancer treatment. No effect of anticancer treatment on either intensive care unit admission rate or respiratory support rate was reported. Chemotherapy, however, resulted in a higher mortality rate of patients affected by hematological malignancies who received chemotherapy within 3 months prior to COVID-19 diagnosis [47].

In line with the previous data, Yekedüz and colleagues’ meta-analysis included 16 studies showing that the administration of immunotherapy, targeted therapy, radiotherapy, and cancer surgery in the last 30 days before the diagnosis of COVID-19 did not increase the risk of severe disease and death in cancer patients [48]. Conversely, although there was no increased risk of severe disease, the risk of death from COVID-19 was higher in the chemotherapy group at the multivariate analysis [48].

A major concern for cancer patients is the potential impact of immune checkpoint inhibition on the clinical course of COVID-19 disease [49]. Immunotherapeutic agents such as programmed death-1 (PD-1) inhibitors, programmed death ligand-1 (PD-L1) inhibitors, or cytotoxic T lymphocyte-associated antigen-4 (CTLA-4) inhibitors act by enhancing T-cell functions against tumor cells or viruses. The adaptive immune cells involved in this process, particularly CD8+ and CD4+ T cells, are crucial for regulating immunity against viruses, thus immune checkpoint inhibitors (ICI) may improve the immunological control of viral infections, potentially offering a protection against the development of severe COVID-19 disease.

Growing evidence demonstrated upregulation of immune checkpoint receptors in severe cases of COVID-19 disease characterized by lymphopenia [50]. Indeed, following SARS-CoV-2 infection, cytokine storms induce T-cell hyperactivation and culminate in T-cell depletion, resulting in lymphopenia [51]. T-cell depletion associated with severe COVID-19 can lead to acute respiratory distress syndrome (ARDS). As ICI increases both the number and function of cytotoxic T cells, the clinical features of COVID-19 could make ICI a considerable option for treating COVID-19 disease.

Immunotherapies (convalescent plasma therapy, human monoclonal antibodies, and interferon) have been shown to be safe and effective treatment options for COVID-19-positive patients [52]. Immune checkpoint receptors studied as therapeutic targets for anti-COVID therapies include PD-1 as well as novel receptors like NKG2A and C5aR. Preclinical studies have shown that the inhibition of these immune checkpoint receptors enhances T-cell expansion and antitumor immunity [53]. Particularly, the inhibition of NKG2A increases the anti-tumor activity of T-cells and NK cells [54].

Despite these evidences, some concerns remain on the application of ICI for the treatment of COVID-19. T-cells reactivated by immunotherapy may increase cytokine secretion as well as the risk of a cytokine storm, worsening the course of COVID-19 disease and leading to unfavorable organ injury. The use of the IL-6R inhibitor tocilizumab has been shown to reduce cytokine release, allowing patients to continue receiving immunotherapy [55].

Rogiers and colleagues conducted a multicenter, retrospective cohort study of 110 patients with SARS-CoV-2, contracted during treatment with ICI, with the aim of identifying risk factors for hospitalization and mortality [56]. Thirty-two percent of patients were admitted to hospital. Factors associated with an increased risk of hospitalization were ECOG ≥ 2, combined ICI treatment, and the presence of symptomatic COVID-19 disease. Seventy-three percent of patients either discontinued or stopped ICI due to SARS-CoV-2 infection. The mortality rate was 16% and was related to COVID-19 in almost half of the cases (7%). All patients who died had advanced disease and only four had been admitted to the intensive care unit. COVID-19-related mortality in the ICI-treated population was higher than that reported in the general COVID-19 positive population (1.4–2.3%) but was on the lower side of the range reported for cancer patients (7.6–33%) [56].

Another challenge for lung cancer patients receiving ICI-based therapy during the COVID-19 pandemic is the differential diagnosis between ICI-induced pneumonia and COVID-19-related pneumonia. Indeed, there is a wide range of radiological features and clinical symptoms that overlap, sometimes, complicating the clinical management of these patients.

## 6. SARS-CoV-2 Vaccination: The Real Turning Point

The SARS-CoV-2 vaccine has recently emerged as the main weapon to fight the COVID-19 pandemic.

The various candidate anti-SARS-CoV-2 vaccines can be grouped according to the technological platform used to their development in order to elicit a protective immune response [57]. Two vaccines based on SARS-CoV-2 attenuated viruses have reached the clinical trial phase [58]. Vaccines based on inactivated SARS-CoV-2 viruses are more stable than live attenuated ones but their limitation is mainly linked to the short duration of elicited immune memory [57]. Vaccines based on SARS-CoV-2 proteins are mostly produced in vitro using recombinant DNA technology. Some of them are currently in phase II clinical trials. No vaccines based on naked DNA have been registered for human use yet. There are different vaccines based on mRNA carried by liposomes and some of these have recently completed phase III studies. The antigen encoded by the mRNA is represented by the Spike protein, its variants, or its fragments, while the DNA encoding for the Spike protein can be transported into cells by viral vectors [57]. The majority of vaccines based on viral vectors technology are currently under investigation in phase III trials [58].

Following the recommendation of the European Medicines Agency (EMA), on 21 December 2020, the European Commission authorized the first vaccine against COVID-19, mRNA BNT162b2 (Comirnaty), manufactured by Pfizer and BioNTech. There are limited data on the efficacy of vaccination in cancer patients, with most of the available information relating to the influenza vaccine [59]. Observational clinical studies have shown lower influenza mortality and morbidity in vaccinated cancer patients, suggesting an effective immune response, even during systemic chemotherapy [60,61,62]. Data on the efficacy and safety of SARS-CoV-2 vaccines in cancer patients are still limited, as these patients were largely excluded from pivotal trials. Only two studies enrolled cancer patients (4% and 0.5% of the total population, respectively), but no dedicated analysis was performed [63,64].

Recent published data suggested a satisfactory serological status following the second dose of vaccine in cancer patients [65,66,67,68,69].

An observational, prospective cohort study analyzed the humoral immune response to SARS-CoV-2 vaccination in 232 cancer patients compared to 100 healthy volunteers [70]. Two weeks after the second dose of vaccination, 90.5% of cancer patients were seropositive compared to 98% of healthy controls (*p* = 0.015). The majority of seronegative patients were male, older than 70 years, with comorbidities that were on active anticancer treatment. The median antibody titers of cancer patients were significantly lower than controls. An interesting finding concerning the role of cigarette smoking in the immunological response of cancer patients showed a significant association between antibody titer and smoking status, with significantly higher values found in the non-smoker compared with the smoker cohort (*p* = 0.006) [70]. Similarly, an Israeli prospective study showed a 90% seroconversion rate in 102 cancer patients undergoing active anticancer treatment compared to 100% in 78 healthy controls following the second dose of BNT162b2 vaccine [65]. Lung cancer patients had a 92% seroconversion rate [65]. A follow-up analysis found that 5.5 weeks after the second dose, 100% of healthy controls (78/78) vs 90% of cancer patients (90/102) were seropositive, with a significantly lower median IgG titer than controls [71]. In a larger study including a total of 200 cancer patients who received full dosing with one of the Food and Drug Administration-approved vaccines, relatively lower IgG titers were observed after vaccination with adenoviral (Ad26.COV2.S) compared to mRNA-based vaccines (mRNA-1273, BNT162b2) [72]. Chemotherapy has been associated with lower seropositivity rates than both targeted and immunotherapy treatments [73].

Vaccines could potentially be less effective in lung cancer patients than healthy controls. In the Israeli study, patients with thoracic malignancies had lower antibody titers (1334 AU/mL) than both healthy subjects (7160 AU/mL) and the overall cancer cohort (1931 AU/mL) [65]. Recently, Gounant and colleagues reported the results of the prospective observational COVIDVAC-OH study investigating the efficacy of vaccination against SARS-CoV-2 in patients with thoracic malignancies [74]. Only seven cases of COVID-19 were observed among 306 vaccinated subjects (2.3%), confirming the efficacy of COVID-19 mRNA vaccines in lung cancer patients. Of 269 serological tests available 2 weeks after the second vaccine dose, 6.3% were still negative (<50 AU/mL). Lack of immunization was associated with age and chronic corticosteroid treatment. Thirty patients with persistent low antibody titers achieved an immunization rate of 88% after receiving a third vaccine dose [74].

To date, February 2022, more than 9 billion doses of vaccine have been administered worldwide, with the rate of administration varying greatly between different areas of the world. Month after month the rate of positivity, hospitalizations and mortality from SARS-CoV-2 significantly decreases over the time, allowing the reopening of major services and borders.

## 7. Conclusions

SARS-CoV-2 infection has dramatically impacted the real-world management of cancer patients. Given the higher risk of a longer and more severe course of COVID-19 disease in lung cancer patients, oncological services have been profoundly reorganized. The world’s leading professional organizations provided new recommendations for the diagnosis, treatment, and follow-up of lung cancer patients during the pandemic. Telemedicine was preferred for non-urgent visits, and screening programs were temporarily suspended, leading to possible diagnostic delays and an estimated increase of cause-specific mortality. The vaccination campaign has definitively inverted this negative trend, with the administration of the booster dose prioritized in frail immune-depressed patients. The efficacy and duration of a humoral immune response in cancer patients still represents an opened question, requiring further investigation in dedicated studies.

This work aimed to identify the main characteristics of SARS-CoV-2 infection in lung cancer patients in terms of transmission, pathophysiology, risk assessment, and outcomes, addressing clinical needs related to the management of patients with thoracic malignancies during a pandemic. When community transmission is controlled, further studies will be needed to monitor trends in large cohorts of cancer patients over the time.

## Figures and Tables

**Table 1 biomedicines-10-00776-t001:** ESMO recommendations for the clinical management of lung cancer patients in the COVID-19 era.

	High Priority	Medium Priority	Low Priority
Outpatients	New diagnosis of stage ≥ II invasive LC with disease-related symptoms	New diagnosis of stage I LC	
		Follow-up for pts at high risk of relapse	Follow-up for pts at low/intermediate risk of relapse
	Visits for treatment administration	Symptoms from treatment: convert to telemedicine visits where possible	Psychological support: convert to telemedicine
Surgery	Surgery for T2N0, resectable T3/T4, resectable N-1/N2	Surgery for T1AN0	Surgery for pure GGO nodule (T1a)
	Drainage ± pleurodesis of pleural and/or pericardial effusion		
Early Stage LC	Adjuvant CT in T3/4 or N2, young and fit pts	Adjuvant CT in T2b-T3N0 or N1	Adjuvant CT for stage T1A-T2bN0 with negative prognostic features
			Adjuvant CT for elderly or pts with comorbidities should be omitted
	Neoadjuvant CT for stage II		
	Concomitant CT-RT for SCLC stage I/II		
Locally Advanced LC	Neoadjuvant CT for NSCLC stage III		
	Concomitant CT-RT for SCLC or unresectable NSCLC stage III		
	Starting consolidation durvalumab (within 42 days)		
Metastatic LC	1st-line treatment	Consider oral CT instead of intravenous	Postpone antiresorptive therapy
	Start 2nd-line treatment in symptomatic and progressive disease pts	Start 2nd and later lines treatment in asymptomatic pts	
	Anti-PD-(L)1 scheduled cycles may be modified/delayed using 4- or 6-weekly dosing	For pts ongoing with IO from more than 12/18 months, consider enlarging intervals	Consider discontinuation of IO after 2 years of treatment
Radiation	Unresectable stage II-III not feasible for CT	SABR-SBRT for stage I	Palliative RT for symptomatic patients (e.g., bone or chest pain)
	Life-threatening conditions (e.g., SVC obstruction, hemoptysis, spinal cord compression)	Adjuvant PORT for R1	Adjuvant PORT for N2 R0
		PCI in limited stage SCLC	PCI in extensive stage SCLC may be replaced by MRI active surveillance

## Data Availability

No new data were created or analyzed in this study. Data sharing is not applicable to this article.

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
