# Peer review of "COVID-19 and Lung Cancer: A Comprehensive Overview from Outbreak to Recovery"

_biomedicines, 2022, doi:10.3390/biomedicines10040776_

Round 1

Reviewer 1 Report

The efforts of the authors are highly appreciated. Suggestions to improve the manuscript are shown below:

  1. Include a flowchart for your study.
  2. List causes of the outbreak in tabular form or pie chart and discuss.
  3. Summarize recent studies on Covid-19 related to lung cancer.
  4. Summarize benefits and limitations of various treatments/vaccines.
  5. List the effects of Covid-19 on lungs and discuss the link between Covid-19 and lung cancer.
  6. In the conclusion section, include a subsection on “Implications of study for research, policy and industry”.

  1. Kindly cite related papers listed below, related in terms of Covid-19 and public health:

A.O. Sojobi, K.M. Liew. Multi-objective optimization of high performance bio-inspired prefabricated composites for sustainable and resilient construction. Composite Structures 279 (2022) 1-16

AO Sojobi, T. Zayed. Impact of sewer overflow on public health: A comprehensive scientometric and systematic review. Environmental Research 203 (2022) 1-29

Cavanna L, Citteirio C, Madaro S, Bacchetta N et al. COVID-19 vaccines in adult cancer patients with solid tumours undergoing active treatment: Seropositivity and safety. A prospective observational study in Italy. European Journal of Cancer 157 (2021) 441-449 

Zhu Z, Zheng Z, Liu J. Comparison of COVID-19 and Lung Cancer via Reactive Oxygen Species Signaling. Frontiers in Oncology 11 (2021) 1-15 

Robilotti EV, Babady NE, Mead PA, Rolling T, Perez-Johnston R, Bernades M et al. Determinants of COVID-19 disease severity in patients with cancer. Nature Medicine 26 (2020) 1218–1223 

  1. Revise the abstract based on new results/insights obtained.
  2. Kindly differentiate conventional lung cancer from lung cancer caused by Covid-19. Please include clear pictures for illustration.

Author Response

The efforts of the authors are highly appreciated. Suggestions to improve the manuscript are shown below:

  1. Include a flowchart for your study.

We detailed our search strategy in a dedicated section. A flow chart is not mandatory for this kind of work (literature review).

2. List causes of the outbreak in tabular form or pie chart and discuss.

We discussed the causes of the outbreak in the epidemiology section.

  1. Summarize recent studies on Covid-19 related to lung cancer.

We added and summarized the most recent studies on COVID-19 and lung cancer in the paragraph “SARS-CoV-2 vaccination: the real turning point”

4. Summarize benefits and limitations of various treatments/vaccines.

We have implemented the paragraph “SARS-CoV-2 vaccination: the real turning point” including new data on vaccines.

5. List the effects of Covid-19 on lungs and discuss the link between Covid-19 and lung cancer.

We edited the paragraph “immune-pathophysiology of SARS-CoV-2 lung injury” as requested.

6. In the conclusion section, include a subsection on “Implications of study for research, policy and industry”.

In our opinion the conclusion in its current version already include major implications

7. Kindly cite related papers listed below, related in terms of Covid-19 and public health:

A.O. Sojobi, K.M. Liew. Multi-objective optimization of high performance bio-inspired prefabricated composites for sustainable and resilient construction. Composite Structures 279 (2022) 1-16

AO Sojobi, T. Zayed. Impact of sewer overflow on public health: A comprehensive scientometric and systematic review. Environmental Research 203 (2022) 1-29 

Cavanna L, Citteirio C, Madaro S, Bacchetta N et al. COVID-19 vaccines in adult cancer patients with solid tumours undergoing active treatment: Seropositivity and safety. A prospective observational study in Italy. European Journal of Cancer 157 (2021) 441-449 

Zhu Z, Zheng Z, Liu J. Comparison of COVID-19 and Lung Cancer via Reactive Oxygen Species Signaling. Frontiers in Oncology 11 (2021) 1-15 

Robilotti EV, Babady NE, Mead PA, Rolling T, Perez-Johnston R, Bernades M et al. Determinants of COVID-19 disease severity in patients with cancer. Nature Medicine 26 (2020) 1218–1223 

We quoted and cited some of the suggested articles. In addition we added the following papers:

- Rolfo C, Meshulami N, Russo A, Krammer F, García-Sastre A, Mack PC, Gomez JE, Bhardwaj N, Benyounes A, Sirera R, Moore A, Rohs N, Henschke CI, Yankelevitz D, King J, Shyr Y, Bunn PA Jr, Minna JD, Hirsch FR. Lung Cancer and Severe Acute Respiratory Syndrome Coronavirus 2 Infection: Identifying Important Knowledge Gaps for Investigation. J Thorac Oncol. 2022 Feb;17(2):214-227. doi: 10.1016/j.jtho.2021.11.001. Epub 2021 Nov 10. PMID: 34774792; PMCID: PMC8579698.

- Addeo A, Shah PK, Bordry N, Hudson RD, Albracht B, Di Marco M, Kaklamani V, Dietrich PY, Taylor BS, Simand PF, Patel D, Wang J, Labidi-Galy I, Fertani S, Leach RJ, Sandoval J, Mesa R, Lathrop K, Mach N, Shah DP. Immunogenicity of SARS-CoV-2 messenger RNA vaccines in patients with cancer. Cancer Cell. 2021 Aug 9;39(8):1091-1098.e2. doi: 10.1016/j.ccell.2021.06.009. Epub 2021 Jun 18. PMID: 34214473; PMCID: PMC8218532.

- Linardou H, Spanakis N, Koliou GA, Christopoulou A, Karageorgopoulou S, Alevra N, Vagionas A, Tsoukalas N, Sgourou S, Fountzilas E, Sgouros J, Razis E, Chatzokou D, Lampaki S, Res E, Saridaki Z, Mountzios G, Saroglou G, Fountzilas G. Responses to SARS-CoV-2 Vaccination in Patients with Cancer (ReCOVer Study): A Prospective Cohort Study of the Hellenic Cooperative Oncology Group. Cancers (Basel). 2021 Sep 15;13(18):4621. doi: 10.3390/cancers13184621. PMID: 34572848; PMCID: PMC8466969.

- Eliakim-Raz N, Massarweh A, Stemmer A, Stemmer SM. Durability of Response to SARS-CoV-2 BNT162b2 Vaccination in Patients on Active Anticancer Treatment. JAMA Oncol. 2021 Nov 1;7(11):1716-1718. doi: 10.1001/jamaoncol.2021.4390. PMID: 34379092; PMCID: PMC8358809.

- Thakkar A, Gonzalez-Lugo JD, Goradia N, Gali R, Shapiro LC, Pradhan K, Rahman S, Kim SY, Ko B, Sica RA, Kornblum N, Bachier-Rodriguez L, McCort M, Goel S, Perez-Soler R, Packer S, Sparano J, Gartrell B, Makower D, Goldstein YD, Wolgast L, Verma A, Halmos B. Seroconversion rates following COVID-19 vaccination among patients with cancer. Cancer Cell. 2021 Aug 9;39(8):1081-1090.e2. doi: 10.1016/j.ccell.2021.06.002. Epub 2021 Jun 5. PMID: 34133951; PMCID: PMC8179248.

- Waldhorn I, Holland R, Goshen-Lago T, Shirman Y, Szwarcwort-Cohen M, Reiner-Benaim A, Shachor-Meyouhas Y, Hussein K, Fahoum L, Peer A, Almog R, Shaked Y, Halberthal M, Ben-Aharon I. Six-Month Efficacy and Toxicity Profile of BNT162b2 Vaccine in Cancer Patients with Solid Tumors. Cancer Discov. 2021 Oct;11(10):2430-2435. doi: 10.1158/2159-8290.CD-21-1072. Epub 2021 Sep 2. PMID: 34475136.

- Gounant V, Ferré VM, Soussi G, et al. Efficacy of SARSCoV-2 vaccine in thoracic cancer patients: a prospective study supporting a third dose in patients with minimal serologic response after two vaccine doses. medRxiv. J Thorac Oncol. 2022;17:239–251. https://doi. org/10.1016/j.jtho.2021.10.015.

8. Revise the abstract based on new results/insights obtained.

In our opinion new results/insight included in the revised version of the draft do not require abstract editing

9. Kindly differentiate conventional lung cancer from lung cancer caused by Covid-19. Please include clear pictures for illustration.

We apologize but current available data in the literature do not allow a clear relationship between COVID-19 infection and carcinogenesis to be established. We cited some pre-clinical data, in the absence of validated clinical studies. We were not able to include any picture. 

Reviewer 2 Report

Dear authors,

I have read your manuscript very carefully. Given overview is presented in a very interesting and systematic way.

I have a few questions/comments:

which inclusion/exclusion criteria were relevant to your research (manuscript) – state how you decided to choose research papers you are referring to?

Indicate which keywords you used when searching the literature on the basis of which you decided to choose the papers you are referring to

State which database your search (e.g. PubMed….).

Author Response

Dear authors,

I have read your manuscript very carefully. Given overview is presented in a very interesting and systematic way.

I have a few questions/comments:

which inclusion/exclusion criteria were relevant to your research (manuscript) – state how you decided to choose research papers you are referring to?

Indicate which keywords you used when searching the literature on the basis of which you decided to choose the papers you are referring to

State which database your search (e.g. PubMed….).

Following the reviewer’s comments we included a novel “search strategy” section in the manuscript reporting the required info

Round 2

Reviewer 1 Report

Comments from first review has not been adequately addressed.

Author Response

need more detailed comments